# HIF-Prolyl Hydroxylase Domain Proteins (PHDs) in Cancer—Potential Targets for Anti-Tumor Therapy?

**DOI:** 10.3390/cancers13050988

**Published:** 2021-02-27

**Authors:** Diana Gaete, Diego Rodriguez, Deepika Watts, Sundary Sormendi, Triantafyllos Chavakis, Ben Wielockx

**Affiliations:** Institute of Clinical Chemistry and Laboratory Medicine, Technische Universität Dresden, 01307 Dresden, Germany; Diana.Gaete@ukdd.de (D.G.); diego.rodriguez@mailbox.tu-dresden.de (D.R.); deepika.watts@ukdd.de (D.W.); sundary.sormendi@ukdd.de (S.S.); Triantafyllos.chavakis@ukdd.de (T.C.)

**Keywords:** hypoxia, tumor, PHD, HIF

## Abstract

**Simple Summary:**

In solid tumors, proliferation of cancer cells typically outpaces the growth of functional vessels. The net result is often an obstructed blood circulation and areas of deprived oxygen (hypoxia). To overcome this acute stress, hypoxia inducible factors (HIFs) stimulate the expression of numerous proteins that will support adaptation to this situation and stimulate further growth, differentiation, and even dissemination. The HIF-response is closely controlled by a class of enzymes known as the HIF prolyl hydroxylases (PHDs). They are true oxygen sensors and directly regulate the activity of HIFs. Although many studies are currently focusing on inhibiting the activity of HIFs in tumors, the role of hypoxia signaling is complex and regulating PHDs in a number of tumor settings might be beneficial. This review gives an overview of the literature on the nature of PHDs in tumor-associated cells and discusses available PHD inhibitors and their potential use as an anti-tumor therapy.

**Abstract:**

Solid tumors are typically associated with unbridled proliferation of malignant cells, accompanied by an immature and dysfunctional tumor-associated vascular network. Consequent impairment in transport of nutrients and oxygen eventually leads to a hypoxic environment wherein cells must adapt to survive and overcome these stresses. Hypoxia inducible factors (HIFs) are central transcription factors in the hypoxia response and drive the expression of a vast number of survival genes in cancer cells and in cells in the tumor microenvironment. HIFs are tightly controlled by a class of oxygen sensors, the HIF-prolyl hydroxylase domain proteins (PHDs), which hydroxylate HIFs, thereby marking them for proteasomal degradation. Remarkable and intense research during the past decade has revealed that, contrary to expectations, PHDs are often overexpressed in many tumor types, and that inhibition of PHDs can lead to decreased tumor growth, impaired metastasis, and diminished tumor-associated immune-tolerance. Therefore, PHDs represent an attractive therapeutic target in cancer research. Multiple PHD inhibitors have been developed that were either recently accepted in China as erythropoiesis stimulating agents (ESA) or are currently in phase III trials. We review here the function of HIFs and PHDs in cancer and related therapeutic opportunities.

## 1. Introduction

An expanding tumor mass is characterized by a hypoxic tumor microenvironment because oxygen levels drop as the tumor outgrows the supply capabilities of the surrounding blood vessels. Therefore, hypoxia is a major hallmark of solid tumors. Several studies have shown that tumor biology is significantly affected by cancer-related hypoxia, which includes formation of a dysfunctional and disordered vasculature that is typically seen in fast-growing tumors [1]. Additionally, although extreme hypoxia classically results in cell death in normal cells, this stress can induce changes that enable tumor cells to adapt to and survive in a hypoxic microenvironment. Such a response to deprived oxygen comprises both genomic and transcriptomic changes that may lead to genetic instability, cell cycle arrest, cell death, and differentiation [2]. Eventually, persistent hypoxia exerts a selection pressure that results in the survival of certain tumor cell subpopulations that are capable of growth, invasion, and even metastasis [3,4,5]. This efficient cellular adaptation to variations in oxygen levels is tightly regulated by the hypoxia-inducible factor (HIF) family of transcription factors, which are heterodimeric proteins composed of an oxygen-sensitive alpha subunit (mainly HIF-1α and HIF-2α) and a constitutively expressed beta subunit (HIFβ/ARNT).

Although HIF-1α and HIF-2α share overlapping target genes, both also regulate a set of unique targets that are implicated in unrelated processes, and interestingly, they may display even opposite effects, as recently shown in endothelial cells [6]. Notably, these hypoxia-dependent, HIF-1α- and HIF-2α-induced genes play important roles in regulating different aspects of tumor biology, such as angiogenesis [7], survival [8], proliferation [9], immune system resistance [10], tumor cell plasticity [11], invasion and metastasis [12], chemo- and radio-resistance [13,14], pH regulation and metabolism [15], and maintenance of cancer stem cells (CSCs) [16]. Normoxic conditions do not require HIF activity and they are marked for degradation when the HIFα subunits are hydroxylated at two specific proline residues by specific enzymes, i.e., the prolyl-4-hydroxylase domain (PHD) proteins. PHDs can hydroxylate these proline residues on the oxygen-dependent degradation domain (ODDD) at N- or C-termini (NODDD and CODDD, respectively) of HIF-1α and HIF-2α, which then serves as a signal for HIFα degradation by the oxygen-dependent von Hippel-Lindau (VHL) via the 26S proteasome proteolytic pathway [17,18].

There are three known PHD isoforms— PHD1, PHD2, and PHD3, which are encoded by *EGLN2*, *EGLN1,* and *EGLN3*, respectively, and they have been shown to selectively hydroxylate HIFα subunits. Under normoxic conditions, PHD1 and PHD2 preferentially target HIF-2α and HIF-1α, respectively, while HIF-2α is the preferred substrate of PHD3 under hypoxic conditions [19,20]. Due to its association with various physiological and pathological processes, PHD2 is thought to be the main regulator of this hypoxia pathway (previously reviewed by our group in [21]). Mechanistically, when pO_2_ decreases to levels that inactivate PHDs, HIF-1α, and HIF-2α can no longer be hydroxylated, resulting in their accumulation in the cytosol. Subsequent nuclear mobilization enables their dimerization with the HIFβ subunit and transcription initiation [22,23]. Importantly, regulation of HIF-1/2α by PHDs has been linked to contrasting tumor outcomes (http://www.cbioportal.org/, accessed on 1 April 2020). Another important HIFα regulator is factor inhibiting HIF (FIH), which specifically hydroxylates the asparagine 803 in both HIF-1α and HIF-2α [24,25]. This post-translational modification results in failure of association between HIFα with the adaptor protein p300, crucial for nuclear translocation. Although FIH and PHDs share enzymatic features in HIFα regulation [26], in this review, we will focus on the impact of PHDs and HIFs in cancer and discuss current and potential therapeutic approaches.

## 2. PHDs as Central Regulators of Tumor Development

Our group has previously reported a clear pattern of pro- and anti-tumor effects of PHDs among human cancer types [21,27]. These differences point to the presence of a case-by-case scenario, where the individual PHDs can be either beneficial or detrimental to tumor growth, and thus, potentially define future therapy decisions. Interestingly, more cases have been reported that show over-expressing of PHDs in tumor tissue versus healthy neighboring tissue, with few exceptions [21].

The function of PHD1 during tumor initiation and development has not been extensively studied. This might also suggest that modulation of its expression has only limited impact. Indeed, apart from the substantial amount of cases demonstrating differential expression of PHD2 and PHD3 in human cancers, *PHD1* expression in cancer tissue is more unchanged versus healthy tissue (Table 2 in [21]). Nevertheless, PHD1 has been suggested to operate as an oncogene in triple negative breast carcinoma [28] and prostate cancer [29].

In colorectal cancer (CRC), PHD2 has been associated with a protective role. Through its regulatory subunit B55α, PP2A dephosphorylates PHD2 at Ser125, rendering it non-functional, and consequent accumulation of HIF-1α leads to CRC cell survival in hypoxia through autophagy. Targeting B55α impairs CRC neoplastic growth in vitro and in mice in a PHD2-dependent manner [30]. Similarly, another study in breast carcinoma xenografts reported that, when subjected to a glycolysis inhibitor 2-DG (2-deoxy-glucose) to mimic glucose starvation, tumors that lacked PHD2 showed greater resistance to treatment compared to controls, strongly suggesting that PHD2-mediated B55α degradation facilitates breast cancer cell death in response to chronic glucose deprivation [31]. Alongside the evidence that PHD2 overexpression can be favorable in restricting tumor development, contrastingly, silencing of PHD2 reduces tumor growth and survival in many studies. As shown previously by our group, ablation of PHD2 in different murine tumor cell lines such as Lewis lung carcinoma (LLC) model, B16 melanoma, and LM8 osteosarcoma, led to a significant increase in tumor vasculature, followed by a significant reduction in tumor growth due to enhanced MMP activity and TGF-β release within the tumor microenvironment (TME) [27,32]. Another study showed that PHD2 knockdown in MDA-MB-231 xenografts resulted in significantly lower epidermal growth factor receptor (EGFR) expression levels compared to controls. Nonetheless, the authors claimed that EGFR downregulation was independent of the influence of HIF-1α or HIF-2α [33]. The pro-oncogenic adaptor protein, CIN85 has been recently identified as an indirect regulator of PHD2 activity. Kozlova and colleagues have shown that disruption of the CIN85/PHD2 interaction using CRISPR/Cas9 editing not only led to lower levels of HIF-1α and HIF-2α, but also to significantly impaired tumor growth and migration in a breast carcinoma model (MDA-MB-231) [34]. The group of Vidimar explored the redox properties of a ruthenium organometallic compound (RDC11) that directly interacts with PHD2 and showed that RDC11 reduced HIF-1α protein level and function by promoting the enzymatic activity of PHD2. Upon RDC11 administration in human colorectal adenocarcinoma (HCT116 cell line) in vivo, levels of HIF-1α were significantly reduced and, consequently, VEGF levels and angiogenesis, leading to a reduction in tumor size [35]. Using a human LM2 xenograft model, Koyama et al. [36] investigated subsequent tumor vessel normalization after PHD inhibition using DMOG and showed that tumor vessel normalization was accompanied by angiogenesis, which rescued sensitivity to chemotherapy [36].

Remarkably, although PHD3 also displays pro-tumoral activity, a number of human- and mouse-associated tumors show reduced amounts of PHD3 compared to adjacent healthy tissue. In a lung carcinoma model, PHD3 also exerted tumor-suppressive activity, apart from regulating epithelial-to-mesenchymal transition (EMT), metastasis, and resistance to therapy. PHD3 knockdown in other cell lines (A549 and H1299 cells) enhanced pulmonary metastasis in a HIF-dependent manner that involved upregulation of TGFα, an EGFR ligand [37]. In gastric cancer, cell migration and invasion were significantly higher in PHD3-silenced tumor cells than controls, and both HIF-1 and VEGF showed greater expression [38]. In mouse LM8 osteosarcoma, we showed that PHD3 is a tumor suppressor as silencing of this oxygen sensor led to enhanced tumor growth and dramatically changed vessel morphology that was directly related to significantly activated platelet-derived growth factor (PDGF)-C signaling in the vasculature of PHD3 knockdown tumors [39]. Thus, the impact of the PHDs in tumor progression is diverse and cell-dependent, i.e., tumor cell vs. TME. Therefore, an effective therapeutic approach will require genomic profiling of tumors to identify the correct treatment needs for each patient [40].

## 3. Hypoxia Signaling in the Tumor Microenvironment

The tumor microenvironment (TME) is an ensemble of cancer cells, cancer-associated fibroblasts (CAFs), and immune cells, including pro-tumoral regulatory T (Treg) cells and tumor associated myeloid cells. The development of a dysfunctional tumor vasculature features the TEM as a hypoxic environment. This lack of oxygen dampens PHD-dependent negative regulation of HIFs, and its consequent stabilization launching an array of processes that facilitate cell survival (Figure 1). Within the TME, cell adaptation and selective pressures, such as hypoxia, acidosis [41], competition for space and nutrients [42,43], cooperation, and predation by the immune system [44,45] result in the “survival of the fittest”, wherein those tumor cells that are capable of adapting to such harsh conditions maintain their proliferation and can even disseminate [46,47]. Furthermore, in solid tumors, the dysfunctional sprouting of new vessels [48] and inefficient vascular mimicry [49] favor tumor progression, tumor cell motility, invasion, and metastasis [50,51]. Of all these aforementioned processes, hypoxia remains a central mechanism that aids tumorigenesis, progression, and resistance to chemo- and radiotherapy [52,53,54,55]. Moreover, it is well established that vascular disarray represents a major hurdle in cancer treatment as it impairs delivery of drugs [56,57,58]. Considering this, several authors focused their studies on the role hypoxia pathway proteins might play in tumor-associated vasculature. A decade ago, Loinard and colleagues showed that PDH silencing promotes therapeutic revascularization via VEGF-A and eNOS in a HIF-1α-dependent manner [59]. Accordingly, a more recent study revealed that pharmacological PHD inhibition (using roxadustat) (described in detail later in this review) led to HIF-1α-dependent VEGF activation, and consequent enhancement of vascular coverage [60].

Recently, MacLauchlan and colleagues showed that PHD inhibition by dimethyloxaloylglycine (DMOG) in NIH3T3 cells decreases thrombospondin-2 (TSP2), a potent inhibitor of angiogenesis, in a HIF-1α-dependent manner [61]. Although hypoxia decreases TSP2 transcription levels, a direct regulation by HIF-1α was not reported. Seemingly, TSP2 reduced expression relies on its NO-sensitivity and, therefore, on HIF-1α stabilization [62,63], which has been shown to be dependent on a NO feedback loop [64].

Interestingly, Mazzone et al. showed that heterozygous expression of PHD2 lead to normalization of tumor-associated vasculature, which improved perfusion and oxygenation. Although growth of primary tumor was not affected, normalization factors produced by of endothelial cells (EC), such as VEGF and Flt1 prevent tumor dissemination and metastasis [65]. The same group addressed the use of targeting PHD2 as a potential approach to improve the response to chemotherapy. This study showed that reduction of PHD2 improves vessel perfusion and cisplatin delivery [66]. In line with this, a more recent study showed that superoxide dismutase (SOD3) dependent HIF-2α stabilization (due to decreased PHD2 expression) and perivascular NO accumulation led to enhanced expression of vascular endothelial cadherin (VEC). The subsequent decreased in vessel leakage improved drug delivery of doxorubicin to the tumor site and effectiveness of treatment of primary CRC tumors [67]. Interestingly, this SOD3-HIF-2α-dependent effect also led to increased EC LAMA4α expression (via the Wnt pathway), which resulted in improved effective adoptive transfer of tumor specific CD8 T cells in CRC [68]. In summary, specific targeting of PHDs in vascular endothelial cells appears as promising means to improve the efficacy of chemo- and immunotherapy.

## 4. Tumor Hypoxia Signaling and Metabolism

Rapid growth of tumor cells with concomitant ineffective vascularization lead to an unequal distribution of both oxygen and nutrients, and this added selective pressure promotes an evolutionary metabolic shift in malignant cells to meet the needs of tumor development. A major determinant of cell survival in this toxic environment is the ability to switch from an oxidative metabolism to a glycolytic one, and tolerate the resultant increase in the level of acidosis due to lactate production (reviewed in [69]). As mentioned previously, stabilization of HIF-1α plays a major role in the activation of genes needed to increase angiogenesis, glycolytic metabolism, pH regulation, autophagy, migration, and invasion, and serves to further increase resistance to radiotherapy and chemotherapy [70,71]. In malignant cells, a metabolic shift to fulfil the demands of rapid and uncontrolled growth includes reducing the synthesis of acetyl-CoA from glucose, downregulating fatty acid synthesis [72] and controlling β-oxidation using adipocyte-derived lipids to reduce cell dependence on de novo lipogenesis [73]. Moreover, the switch towards lactate generation from glucose, even under aerobic conditions (referred to as the Warburg effect and reviewed in depth in [74]), is an adaptation to intermittent hypoxia in pre-malignant lesions [75]. Interestingly, accruing evidence shows that tumor cells remain heterogeneous within the same neoplastic mass (intra-tumor heterogeneity), which also contributes to treatment resistance and cancer progression [76,77]. Additionally, the effects of the TME are beneficial for the neoplastic cells as they promote cooperation among tumor stroma cells to favor tumor progression. In that respect, CAFs promote tumor growth and invasion, and they are susceptible to a shift towards a catabolic metabolism because of the hypoxic TME [78]. Zhang et al. have demonstrated that CAFs are predisposed to switch from oxidative phosphorylation to aerobic glycolysis in a HIF-1α-dependent manner to ensure the tumor-promoting effects of CAFs during hypoxia. Reduced isocitrate dehydrogenase 3 complex (IDH3a), accompanied by a decrease in α-ketoglutarate (α-KG), affects the ratio of fumarate and succinate, resulting in HIF-1α protein destabilization through PHD2 activity [79]. In contrast, overexpression of IDH3a impedes the fibroblasts-to-CAFs transformation [78]. Taken together, PHDs appear to play a negative role in the development of CAFs and their recruitment by the TME. Other cell types present in the TME, such as immune cells (Reviewed in [80]), are also susceptible to metabolic derangement to adapt to the harsh conditions seen in the tumor.

## 5. Tumor Hypoxia Signaling and Recruitment/Activation of Immune Cells

The TME actively releases pro inflammatory cytokines, such as TNF, IL-1, and GM-CSF, and cancer cells add IL-6/8 to the mix, further attracting immune cells [81]. Additionally, hypoxia can enhance or reduce, as the case may be, the infiltration of a substantial number of immunosuppressive cells, such as tumor-associated macrophages (TAMs), myeloid-derived suppressor cells (MDSCs), and regulatory T-cells (Treg), as described below (Figure 2).

TAMs have been linked to enhanced tumor vascularization, greater invasion and metastasis, immune tolerance, and tumor chemo-resistance [82]. Lower oxygen concentrations in tumors have been shown as the mechanism underlying both monocyte recruitment and their subsequent differentiation into a pro-tumoral M2 or TAM phenotype [83,84]; in contrast to the pro-inflammatory M1 macrophages. It has also been suggested that hypoxia can dictate the metabolic profiles associated with M1- and M2-polarised cells. Whereas M1 macrophages produce high levels of iNOS [85], show enhanced HIF-1α activity and, thereby, favor glycolysis [86], M2 macrophages are essentially anti-inflammatory, pro-metastatic [87], and produce high levels of Arginase I (Arg1) [85,88] associated with HIF-2α activity [89]. Further, M2 macrophages mainly produce ATP through the oxidative TCA cycle linked to oxidative phosphorylation (OXPHOS) and rely on fatty acid oxidation (or β-oxidation) and glutamine metabolism, which fuels the TCA cycle [86]. For this reason, hypoxia-induced TAMs polarization is considered a major setback in cancer therapy.

The involvement of PHDs in TAM accumulation, polarization and survival has been suggested, and in a recent study, Wang et al. have demonstrated that PHD2 overexpression in murine colon cancer xenografts (CT26 and MC38) decreased tumor burden, M2-TAM infiltration, and levels of inflammatory cytokines, namely, TNF, G-CSF, IL-8, IL-4, IL-1β, and IL-6 [90]. Similarly, another study that used bone marrow derived macrophages (BMDMs) isolated from mice deficient in PHD2 in myeloid cells has shown a role for PHD2 in macrophage activity. Although HIF-1α and HIF-2α are known to modulate macrophage polarization, in this study PHD2 knockout macrophages did not display a clear change in polarization compared to control cells. Moreover, the O_2_ consumption rate (OCR) of the BMDMs was significantly reduced, whilst showing an increased level of extracellular acidic rate (ECAR). These observations underscore the occurrence of a metabolic shift that resulted in lower phagocytosis and migration of the PHD2 cKO macrophages, but not necessarily in changes in polarization [91]. Hence, the fact that these PHD2cKO macrophages affected phagocytosis despite being highly glycolytic might indicate that the metabolic reprogramming itself is not essential, but rather having an active mitochondrial program that provides enough energy for the phagocytic process [92].

Two major categories of myeloid-derived suppressor cells (MDSCs) have been identified in mice, viz., polymorphonuclear CD11b^+^Ly6G^+^Ly6C^lo^ (PMN-MDSCs) and monocytic CD11b^+^Ly6G^-^Ly6C^hi^ (M-MDSCs). There is substantial functional overlap of PMN-MDSCs with tumor-associated neutrophils (TAN)-2 promoting tumor growth [93,94], as opposed to TAN-1 that have anti-tumor activities [95,96]. MDSCs are known to exert very fundamental immunosuppressive functions, such as inhibition of T cell cytotoxicity [97,98], but tumor hypoxia plays a pivotal role in MDSC recruitment [99]. Moreover, HIFs have been suggested to promote the expression and regulation of Arg1 and iNOS [100,101,102], while the Wang and colleagues also documented an anti-inflammatory effect of PHD2, apart from recruiting MDSCs during tumor progression [90]. Specifically, overexpression of PHD2 impaired MDSC recruitment due to a decrease in NF-κB activity that resulted in lower TNF and G-CSF expression, which are crucial cytokines for MDSC mobilization [103,104] from colon cancer cells [90].

Treg-mediated immunosuppression in cancer enables malignant cells to escape detection by host immune system surveillance mechanisms and several reports have confirmed Treg accumulation within the TME [105,106,107] (reviewed in depth in [108]). Moreover, a hypoxic environment increases HIF-1α-induced expression of the distinct Treg marker and master regulator forkhead box P3 (Foxp3) [109,110]. In contrast, PHD2 has been recently reported to modulate immunosuppressive capabilities of the Tregs. For example, Yamamoto and colleagues have reported that silencing of PHD2 using doxycycline (DOX)-induced expression of shRNAs for PHD2 stabilized HIF-2α in the hematopoietic compartment, which resulted in the loss of immunosuppressive function in Tregs. Moreover, the Treg population associated with a naïve phenotype (CD44loCD62Lhi) was significantly reduced, while the effector memory cell (CD44hiCD62Llo) population was increased [111]. This clear connection between PHD2 and Treg function warrants further studies that explore the role of PHD2 in TME-associated immunosuppression and targeting of PHD2 could potentially lead to loss of tumor-induced immune tolerance, and hence, more efficient immunosurveillance. Additionally, PHD3 is crucial for the development of Tregs, as anti-PHD3 siRNA downregulated Foxp3 and upregulated HIF-1α expression, leading to development of Th17 cells [112].

## 6. Hypoxia Signaling in Cancer Stem Cells (CSC) and the Epithelial-to-Mesenchymal Transition (EMT)

Of the many features of CSCs, the most fundamental are enhanced DNA-repair mechanisms and induction of a quiescent state [113]. As conventional therapies primarily target highly dividing cells, quiescent CSCs represent a dangerous subpopulation that remains undetected and, more importantly, unaffected. Furthermore, inefficient oxygen distribution throughout the tumor allows undifferentiated cells to populate the hypoxic region and there is evidence that CSCs can metabolically adapt to using lactate as their energy source during metastatic colonization (Warburg effect) in a HIF-1-dependent manner [114,115,116]. As both HIF-1 and HIF-2 are highly expressed in CSCs [117], the use of HIF inhibitors, in combination with current therapies, can be developed into an effective counter measure to reduce resistance.

Glioblastoma (GBM) is an aggressive but very common brain tumor. The fast-growing nature of GBMs contributes to the development of an acute intratumoral hypoxic microenvironment, resulting in heterogeneity among malignant cells [118,119]. The glioma stem-like cells (GSCs) certainly benefit from the hypoxic environment as they acquire multipotency and self-renewal capacity, both of which are linked to treatment-resistance and tumor recurrence [120,121,122]. Not surprisingly, HIF-1α expression is increased in both GSCs and non-GSCs, and it has been reported that GSCs promote their tumorigenic capacity and expansion in a HIF-1α–dependent manner [123]. Thus, hypoxia-mediated expansion of GSCs has become a potential target for glioblastoma therapy. Additionally, HIF-2α activity has been related to GSCs and tumor progression. A compelling analysis of angiogenesis-related factors in 50 human GBM samples concluded that there was a significant abundance of HIF-2α over HIF-1α [124]. Furthermore, several studies have demonstrated that HIF-2α is preferentially expressed within a tumor stem cell subpopulation, stimulated by CD44 and that it drives tumor differentiation [123,125,126]. Mechanistically, in vivo studies have shown that the intracellular domain (ICD) of CD44 binds to and activates HIF-2α, but not HIF-1α, in an oxygen-independent manner [125] (Figure 2).

A factor that contributes to CSC development is epithelial-to-mesenchymal transition (EMT), which constitutes a highly coordinated program wherein epithelial cell markers are suppressed while mesenchymal markers are upregulated. This program does not work as a simple on/off switch; in fact, EMT markers manifest in varying degrees and cells can also regress to a more epithelial state. Under non-pathological conditions, the EMT program is required for tissue morphogenesis during embryonic development and is coordinated by multiple transcription factors (EMT-TF), including Slug, Snail, Twist, Zeb1, and Zeb2/SIP1. Each of these EMT-TFs is capable of repressing E-cadherin expression, leading to changes in gene expression, including that of mesenchymal markers, and increasing cellular motility. Moreover, cancer cells that have undergone EMT display CD44^high^/CD24^low^ expression, and are characterized by many of the properties seen in self-renewing stem cells. The final outcome of these changes are related to development of resistance to anti-tumor therapies and initiation of tumor growth in secondary organs [126,127,128].

The EMT program can be triggered by a variety of mechanisms, including intra-tumoral hypoxia [129]. HIF-1α can particularly induce EMT by upregulating the expression of EMT-TFs in several types of cancers, including lung, colorectal, and head and neck cancers [130,131,132,133,134]. Besides hypoxia, adaptive changes in cancer cells following therapy (such as the Warburg effect) [135], as well as several growth factors, can trigger EMT programs, with the relevant factors being transforming growth factor beta (TGF-β), receptor tyrosine kinase (RTK) ligands, epidermal growth factor (EGF), insulin growth factor (IGF), hepatocyte growth factor (HGF), fibroblast growth factor (FGF), and platelet-derived growth factor (PGDF) [126,127]. The hypoxia pathway regulates several of these growth factors as well [21,136,137]. Additionally, microRNAs (miRNAs) regulate EMT and the key candidates include the miRNA-200 family, miRNA-205, miRNA-155, let-7, and miRNA-34a [126,138]. Like the growth factors, some miRNAs may be regulated by hypoxia and/or affect the hypoxic response, e.g., miRNA-155, let-7, and miRNA-34a [137,138]. Increasing expression of the microRNA-200 family and Let-7a is used therapeutically, and a MIR34a mimic has been shown to have anti-tumor activities; however, clinical trials were terminated due to immune-related adverse effects [139,140].

As indicated above, targeting CSCs remains challenging because cells that have undergone at least one partial EMT program exhibit intensified resistance to apoptosis or an ability to force out cytotoxic drugs [127]. Therapies that target EMT aim to halt CSC production to hamper metastasis and cancer progression and have focused on three approaches: (1) targeting EMT-inducing signals; (2) reversing EMT; and (3) killing cells in an EMT-like state. A few clinical trials testing the efficacy of suppressing the EMT program are underway, and while Notch or HIF-1α inhibitors have been proposed to work by targeting stemness or the EMT, TGFβ inhibitors have been used to target tumor cells that have activated versions of the EMT program, and the WNT/FZD pathway is targeted for tumor dedifferentiation [127,141]. As EMT is induced by HIF-1α and therapy targets are frequently inhibited, PHDs have not been explored as therapy targets.

## 7. Hypoxia Signaling in Neuro-Endocrine Tumors

The peripheral nervous system is composed of different types of cells located throughout the body and they serve as the origin of many kinds of benign and malignant tumors. Examples include neural crest-derived neuro-endocrine tumors (NETs), such as paragangliomas (PGLs) that originate from extramedullary paraganglia, as well as pheochromocytomas (PCCs), which are endocrine tumors arising from chromaffin cells located in the adrenal medulla [142,143]. Neuroendocrine properties of these tumors lead to excessive production of catecholamines, such as dopamine, norepinephrine, and epinephrine [144].

PCCs and PGLs are currently subdivided into two major clusters based on underlying mutations in the predisposing genes: the pseudohypoxia-associated cluster 1 and the kinase signaling-associated cluster 2; however, a potential third cluster associated with WNT-signaling has also been recently described [145,146]. Cluster 1 includes tumors associated with mutations in *VHL*, succinate dehydrogenase (*SDHx*) genes or *PHD2*, which lead to stabilization of HIF proteins, especially HIF-2α, thereby creating a pseudohypoxic state [144,147]. Additionally, gain-of-function mutations in exon 9 and 12 of HIF-2α have been added to the list of genes associated with PCCs and PGLs [148,149]. These mutations in HIF-2α result in defective proline residues at the hydroxylation sites, resulting in reduced degradation, and hence, their stabilization. As mentioned before, activation of the HIF pathway also facilitates the Warburg effect, which favors tumor growth by overexpressing genes involved in the glucose metabolism [150]. Another important factor that is upregulated in cluster 1 associated tumors, specifically in relation to SDH and VHL mutations, is mir-210. Its expression is induced by HIF-1α and it is believed to regulate the expression and function of tumor-associated genes [151].

Additionally, HIF-2α stabilization due to mutations in any of the above-mentioned genes in cluster 1 PCCs and PGLs leads to diminished transcription of Phenylethanolamine N-methyltransferase (PNMT), which is the central enzyme that regulates the conversion of norepinephrine to epinephrine. Even though a majority of these tumors are benign, 15–20% metastasize; however, in the absence of markers to distinguish between the two, development of appropriate treatment strategies is essential. As it is well established that HIF-2α is a major driver of PCCs and PGLs, therapeutic targeting of HIF-2α is a potential treatment strategy. However, targeting using small molecules only came to light once the structure of the HIF-2α/HIFβ dimer was resolved by crystallography, and this led to the identification of a large protein cavity in the HIF-2α PAS-B domain. Both in vitro and in vivo models of these rare neuroendocrine tumors showed inhibition of tumors by treatment with HIF-2α inhibitors [152]. Therefore, HIF-2α-specific inhibitors represent a successful method of targeting the core of PCCs and PGLs. Nevertheless, further research and clinical trials are necessary to establish any potential treatment strategy using HIF-2α inhibitors in combination with other existing anti-tumor therapies [153].

CSCs have also been suggested as potential tumor therapy targets in PCC and PGL [154], and it is not surprising that cancer cells from cluster 1 pseudohypoxia-related tumors express CSC markers [155]. Targeting CSCs via surface markers or by inhibiting developmental stem cell pathways has been used in the clinic for the treatment of other tumors, such as in the lung [156], and given their promising outcome, CSC targeting might prove useful, even in PCCs and PGLs.

## 8. PHD Inhibitors—Useful as an Anticancer Therapy

In recent years, PHD inhibitors (PHDi) have been developed as erythropoiesis stimulating agents (ESA) for use in patients suffering from anemia that is often associated with kidney disorders [157,158]. Pharmacological inhibition of PHDs leads to HIFα protein stabilization, including HIF-2α in erythropoietin (EPO)producing cells (EPCs), which results in enhanced EPO production, predominantly in the kidney [159]. This hormone then translocates into the bone marrow where it regulates survival and differentiation of erythroid progenitors to stimulate erythrocyte production. Systemic use of these PHDi might obviously also impact other cell lineages in a variety of different organs. Therefore, it is of utmost importance to increase our understanding of the role of these oxygen sensors using animal models involving conditional transgenic mouse models.

Different pharmaceutical laboratories have approached the pharmacological PHD inhibition and its possible impact in tumor development and therapy. The major drugs currently being studied are roxadustat (FG-4592), vadadustat, daprodustat, and molidustat (summarized in Table 1). These compounds have been challenged in different cancer models, resulting in positive and negative outcomes, emphasizing the need for more research in the local and systemic effects that the inhibition of PHDs may have.

Roxadustat (FG-4592) is a 2-OG analog and was developed as an inhibitor of HIF-PHDs by FibroGen, AstraZeneca, and Astellas Pharma [164]. Seeley and colleagues studied its implications in cancer progression and found that in MMTV-Neu^ndl^-YD5 (NeuYD) mice, which are a model of spontaneous mammary tumor development that are sensitive to VEGF. Moreover, oral administration of Roxadustat yielded no differences in tumor development compared to vehicle-treated MMTV mice [167], confirming that, despite HIF stabilization translating to increased erythropoiesis, the compound has no tumor promoting effects in vivo. This result was later challenged by Koyama et al. [36], who compared DMOG and roxadustat as PHD inhibitors in LLC tumor models, and showed clear tumor vessel normalization and rescue of chemotherapy sensitivity in tumor-bearing mice challenged with the compounds [36,162]. A very detrimental effect to consider when HIFs are activated is the increase in glucose uptake and its consumption during glycolysis, which eventually results in enhanced glycogen storage [168]. This allows cells to survive extreme hypoxic conditions, which, during a neoplastic event, can eventually drive adaptation of malignant cells towards cancer progression, invasion, and metastasis [169]. However, whether these effects could potentially favor tumor progression has not yet been studied. Furthermore, roxadustat can also inhibit tumor growth of macrophage-abundant tumors by facilitating the phagocytic function of Ly6C^lo^ tumor-infiltrating macrophages, which, at least in part, contribute to vessel normalization [161].

Vadadustat [163], developed by Akebia Therapeutics, stabilizes both HIF-1α and HIF-2α and has the potential to inhibit all PHD isoforms but with a preference for PHD3 [164]. One of the main concerns with HIF stabilization by PHDs inhibition with Vadadustat is the risk of facilitating tumor progression due to angiogenesis, secondary to increased VEGF expression [170]. Pergola and collaborators have tested this hypothesis and have reported that levels of VEGF in plasma were not affected after vadadustat treatment in a phase 2b clinical study [163]. Further, a recent study by Nishide and colleagues confirmed this in a mouse model of cancer and showed that vadadustat induces tumor normalization and reduces hypoxic regions within tumor tissue. However, when compared to other PHD inhibitors tested simultaneously, these tumors showed enhanced expression of other angiogenesis markers, such as *Notch1*, *eNOS,* and *Hey1*, and a mild increase in pro-inflammatory markers [162].

Daprodustat [171], developed by GlaxoSmithKline, preferentially inhibits PHD1 and PHD3 [164], and both HIF-1α and HIF-2α isoforms stabilize upon treatment, attesting to its efficacy in activating the hypoxia pathway. Importantly, no carcinogenicity potential was detected for this compound even at high pharmacological doses [165]. Daprodustat was also effective in a mouse LLC model as it resulted in better normalization of the tumor vessels with enhanced pericyte coverage that was linked to diminished presence of angiogenic factors. Moreover, tumor growth was significantly reduced compared to untreated tumors [162].

Molidustat [172], developed by Bayer, has a preferential sensitivity for PHD2 [164]. In a report by Nishide et al., this inhibitor also diminished LLC tumor growth that was linked to enhanced blood vessel maturation and an increase in their functionality [162]. Furthermore, Molidustat has been tested in combination with the proliferation inhibitor, gemcitabine, in a mouse model of breast cancer (MDA-MB-231) [173]. In vitro, a dramatic reduction in cell viability was shown in comparison to control, PHD inhibitor alone, or gemcitabine alone. Although the authors reported an increase in VEGF, both in gene expression and protein release into the culture media, it resulted in no significant changes in angiogenesis, other than dramatic anticancer effects in vivo [166].

## 9. Conclusions

This review explored current advances in the biology of PHD enzymes and their association with cancer progression and therapy. The involvement of PHDs in tumor development in many cases may appear paradoxical, because, while on the one hand there is evidence showing that PHDs can be detrimental for hypoxia adaptation and cancer progression, the use of PHD inhibitors leads to lower tumor growth and metastasis by diminishing immune tolerance and increasing tumor vessel normalization. Moreover, recent evidence advocates for the use of combination therapies, including pharmacological targeting of PHDs, to ensure proper targeting of the individual insults generated by malignant cells. More research is required to obtain a better understanding of the complex mechanisms underlying the effects of hypoxia pathway proteins (i.e., HIFs and PHDs) that are involved in many different types of cancers and pathologies.

## Figures and Tables

**Figure 1 cancers-13-00988-f001:**
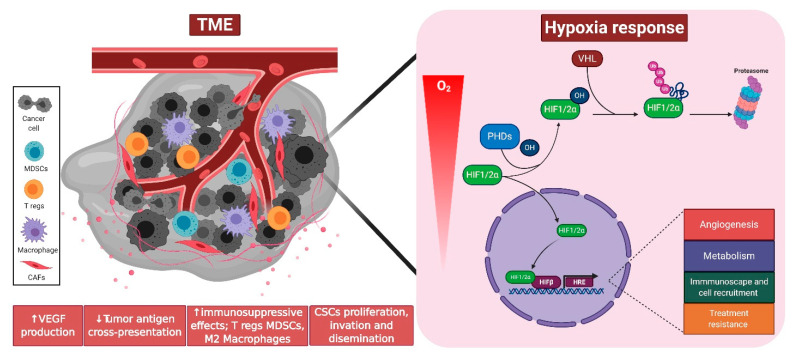
The hypoxic tumor microenvironment (TME) favors hypoxia inducible factor (HIF)-dependent transcriptional responses in cancer and/or stromal cells. “Survival of the fittest” leads to excessive proliferation and dissemination of the more aggressive malignant cells capable of survival under the harshest conditions. For more details, please see text. Developed in Biorender.com.

**Figure 2 cancers-13-00988-f002:**
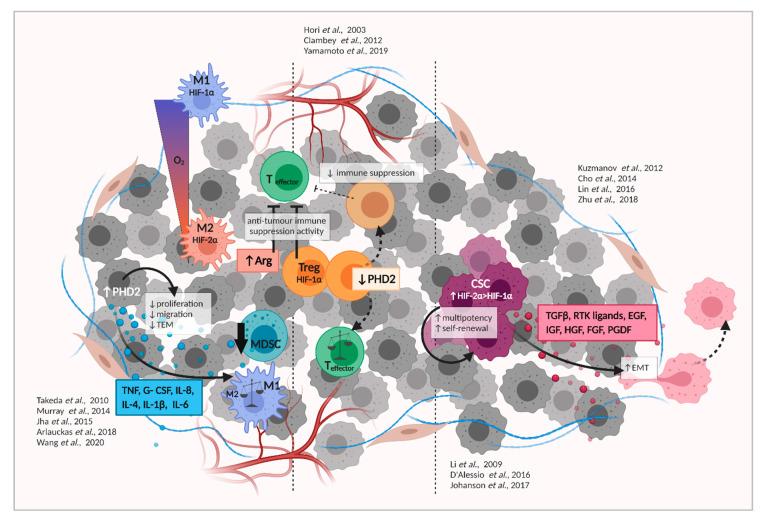
The impact of hypoxia pathway proteins on immune cells and cancer stem cells, and their impact on the development of the tumor. For more details, please see text. Developed in Biorender.com.

**Table 1 cancers-13-00988-t001:** Prolyl hydroxylase domain protein (PHD) inhibitors in cancer-related studies in vivo.

PHDi	Molecular Inhibition	Selected Studies in Cancer Models	
Roxadustat	All HIF-PHDs interactions	- Increased erythropoiesis in MMTV-Neu but no differences in tumor development [160]. - Tumor vessel normalization in mouse LLC tumor, reduced growth [161,162] and sensitivity to chemotherapy [36].
Vadadustat	PHD3 > PHD2 > PHD1	- No increased plasma VEGF in patients [163] and upregulates HIF-2α > HIF-1α [164] - Vessel normalization and reduced tumor growth, but enhanced expression of angiogenesis markers > Rox., Dap., and Mol [161].
Daprodustat	PHD1 > PHD3 > PHD2	- High doses of drugs did not show carcinogenic potential in vivo [165]. - Reduced tumor growth, vascularization, and diminished hypoxic regions [162].
Molidustat	PHD2 > PHD3/PHD1	- In vitro reduced tumor viability. - In vivo impaired tumor growth without altering angiogenesis in an MDA-MB-231 [166], but with enhanced normalization in LLC tumors [162]

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
