# Peer review of "HIF-Prolyl Hydroxylase Domain Proteins (PHDs) in Cancer—Potential Targets for Anti-Tumor Therapy?"

_cancers, 2021, doi:10.3390/cancers13050988_

Round 1
Reviewer 1 Report
No further questions
Reviewer 2 Report
I thank the authors to take into consideration my comments.
This manuscript is a resubmission of an earlier submission. The following is a list of the peer review reports and author responses from that submission.
Round 1
Reviewer 1 Report
This is an interesting and well-written overview by Gaete et al. The authors scrutinized the literature and provide a very good overview on a controversial topic, namely to what degree inhibition of PHDs might be useful in cancer therapy.
I only have a few minor points I would lie the authors to consider:
- The authors do not elaborate on FIH controlling transcriptional activity of HIF complexes; still there are FIH inhibitors on the way, so the authors may at least add a sentence that more than just the abundance of the alpha-subunits controls HIF.
- P3 line 117 which ref. of Zhang is meant in this context?
- P3 line 143 – the authors should mention/discuss that current PHIs favor HIF-2 accumulation which might be problematic with respect to polarization
- P3 likewise 163 – HIF-1 expression and iNOS vs HIF-2 and Arg-1 – I do not believe that there is such a strict association but readers may be confused
- P8 line 357 you might want to mention that increases in hct have been tried to improve tumor oxygenation and thus radiation/chemo therapy – in the way PHIs may affect cancer treatment (also ameliorating tumor anemia).
Author Response
Reviewer 1:
This is an interesting and well-written overview by Gaete et al. The authors scrutinized the literature and provide a very good overview on a controversial topic, namely to what degree inhibition of PHDs might be useful in cancer therapy.
We thank the reviewer for the stimulating words.
I only have a few minor points I would like the authors to consider:
- The authors do not elaborate on FIH controlling transcriptional activity of HIF complexes; still there are FIH inhibitors on the way, so the authors may at least add a sentence that more than just the abundance of the alpha-subunits controls HIF.
We added a small paragraph on FIH in the introduction to complete the oxygen sensors.
- P3 line 117 which ref. of Zhang is meant in this context?
We reformulated the sentence referring to the work of Zhang and colleagues in Cancer Discovery and added the reference to that.
- P3 line 143 – the authors should mention/discuss that current PHIs favor HIF-2 accumulation which might be problematic with respect to polarization.
To our knowledge, there is no concrete data available showing that the current PHDi’s would favor HIF2 accumulation over HIF1 in other cells than renal EPO producing cells (REPCs).
- P3 likewise 163 – HIF-1 expression and iNOS vs HIF-2 and Arg-1 – I do not believe that there is such a strict association but readers may be confused.
We reformulated that in the appropriate section.
- P8 line 357 you might want to mention that increases in hct have been tried to improve tumor oxygenation and thus radiation/chemo therapy – in the way PHIs may affect cancer treatment (also ameliorating tumor anemia).
We have inserted this suggestion in another part of the text in a more general context. We believe it is too early to make a suggestion in that direction. Our group is currently studying the impact PHIs in that context.
Reviewer 2 Report
This is a review manuscript on HIF-prolyl hydroxylase domain proteins (PHDs) as potential targets for anti-tumor therapy. The authors reviewed that 1) tumor hypoxia effects cell metabolism, immune cell recruitment and treatment resistance; 2)functions and mechanism of PHD-HIF in tumor development; 3)pharmacological inhibition of PHDs in cancer treatment.
Here is the commens:
1) Author should introduce the genetic informations of PHDs, and also summary the knocked-in/out functions of PHDs in different tumor models (as a table).
2) In each part of tumor hypoxia functions in tumor immune and tumor treatment resistance, it's better to add a figure of each part to show the role of PHDs played in tumor immune and resistance.
3) Is there any direct evidences of PHDs inhibit PCC/PGL progress? Why you split the hypoxia pathway in PCC/PGLs as a single part?
4) In "the PHD-HIF axis as a central regulator of tumor development" part, the authors summarized the details of functions and mechanisms of PHD2 and PHD3, what about PHD1? In this manuscript, the authors spent lots of space on hypoxia relationship with cancer, other than the PHDs.
5) PHD inhibitors already represented a novel pharmacological treatment of anemia associated with chronic diseases, and late phase clinical trials were ongoing. The author summaried studies on cancer of these new coming drugs. I will suggest to add columns in Table 1 about the clinical trials advances of PHDs inhibitors in different cancer type.
Minor comments:
1) The sentence structure of subtitles should be consistant.
Author Response
Reviewer 2: This is a review manuscript on HIF-prolyl hydroxylase domain proteins (PHDs) as potential targets for anti-tumor therapy. The authors reviewed that 1) tumor hypoxia effects cell metabolism, immune cell recruitment and treatment resistance; 2)functions and mechanism of PHD-HIF in tumor development; 3)pharmacological inhibition of PHDs in cancer treatment.
Here is the comments:
1. Author should introduce the genetic informations of PHDs, and also summary the knocked-in/out functions of PHDs in different tumor models (as a table).
2. In each part of tumor hypoxia functions in tumor immune and tumor treatment resistance, it's better to add a figure of each part to show the role of PHDs played in tumor immune and resistance.
Answer to comments 1&2: We appreciate the first comment and acknowledge the usefullness of such a table. However, based on the second comment from the same reviewer and the limitations related to the length/figures/tables of this review, we decided to develop an overview figure as suggested by the reviewer (new Figure 2)
3. Is there any direct evidences of PHDs inhibit PCC/PGL progress? Why you split the hypoxia pathway in PCC/PGLs as a single part?
Our research group has a particular interest in neuro-endocrine tumors and the impact of hypoxia pathway proteins, based on a running consortium. We are currently studying the impact of the PHD2/HIF2 axis in chromaffin cells and potential aberrations in vivo. Therefore, we decided to include this particular topic in our review as a single chapter.
4. In "the PHD-HIF axis as a central regulator of tumor development" part, the authors summarized the details of functions and mechanisms of PHD2 and PHD3. What about PHD1? In this manuscript, the authors spent lots of space on hypoxia relationship with cancer, other than the PHDs.
We agree with the comment of the reviewer and added a paragraph on the role of PHD1 in cancer.
5. PHD inhibitors already represented a novel pharmacological treatment of anemia associated with chronic diseases, and late phase clinical trials were ongoing. The author summaried studies on cancer of these new coming drugs. I will suggest to add columns in Table 1 about the clinical trials advances of PHDs inhibitors in different cancer type.
We understand the suggestion of the reviewer. Unfortunately, there have not been any clinical trials at this point where the PHD inhibitors are tested in cancer patients. All trials that have been undertaken in that respect relate to mouse experiments. We had added a summary of these results as a column to Table 1 in the original version.
Minor comments:
1.The sentence structure of subtitles should be consistent.
We revisited and edited the subtitles throughout the entire manuscript.
Reviewer 3 Report
This interesting manuscript by Gaete and coworkers reviews the potential pros and cons of HIF-prolyl hydroxylases (PHDs) inhibitors as anti-tumor agents. Since PHDs are negative regulators of hypoxia-induced transcription factors, HIF-1a and HIF-2a, the review revolves around the effect of hypoxia on different activities associated with oncogenesis, such as angiogenesis, metabolism, inflammation, resistance to treatment and cancer stem cells. Afterwards, the influence of hypoxia on pheochromocytomas and paragangliomas is reviewed. Finally, an ambiguous section on the role of the PHD-HIF axis in tumor development is included as a preamble on a brief review of the four pharmacological PHD inhibitors tested in clinical trials in humans (although its effects on oncogenesis were obtained in preclinical models). Overall the review is well written and fairly comprehensive. However, perhaps the most important aspect of the potential role of PHDs is that they can have both anti-tumor and pro-tumor effects. This concept is only clearly evident in the section on the role of the PHD-HIF axis in tumor development. Perhaps the authors could consider moving that section to the beginning of the review.
Major points.
- Many preclinical data suggest that the anti-tumor effect associated with PHD inhibition is a consequence of the "normalization" of the tumor vasculature, which promotes a better function of the blood vessels supplying O2 to the tumor and the reduction of hypoxia. The paradox of why the inhibition of PHDs, enzymes activated by the increase in pO2, improves vascular function is not sufficiently explained in the review. It is simply mentioned (page 3 lines 93-94) the study by Mazzone et al, in which the normalization of the vasculature of tumors implanted in hemizogoto mice for PHD-2 is observed, but this review does not go into the molecular mechanisms of this normalization. In fact, both the study by Mazzone et al, and others published more recently (see PMID: 29422508) indicate that the mechanism behind this normalization is the differential activity of PHDs on HIF-1a and HIF-2a hydroxylation. The low levels of PHD2, in the case of the hemyzogous mouse, or the partial inhibition of PHDs by nitric oxide (PMID: 29422508 and PMID: 17060326) allows the degradation of HIF-1a but the accumulation of HIF-2a, which seems to be a critical transcription factor in the remodeling of blood and lymphatic vessels. Furthermore, the stabilization of HIF-2a has also been associated with greater infiltration of cytotoxic lymphocytes and longer disease-free time in patients with colorectal cancer (PMID: 32591431). These molecular aspects must be discussed in depth so that the reader understands why PHD blockade may exert anti-tumor effects in vivo in some cases.
- Page 4, lines 149-151. From my point of view this sentence should be rewritten, since it seems to indicate that PHD2-KO macrophages do not express any polarization marker. However, in the study by Guentsch et al. what is observed is that there are no differences in the expression of M1 and M2 markers between macrophages WT or PHD2-KO, both under resting conditions (bone marrow isolated) and after stimulation with polarizing agents. It is also important here that the authors discuss in some detail some surprising data obtained on PHD2-KO macrophages. Specifically, the fact that these PHD2-KO macrophages have affected the phagocytosis despite being highly glycolytic, a metabolic process associated with enhanced phagocytosis and polarization towards M1 macrophages (PMID: 31333642). Perhaps this indicates that it is not so important the metabolic reprogramming itself, but rather having an active mitochondrial program that provides enough energy for the phagocytic process.
Minor points:
- Throughout the manuscript. I would appreciate the addition of a hyphen in HIF-1alpha and HIF-2alpha, for consistency with all the literature on this topic.
- 3, line 119. It seems that a reference is missing at the end of that sentence.
- 4, line 131. I think the appropriate expression is Regulatory T-cells
- 8, lines 344-345. The phrase should be rewritten by substituting "oral application" by "oral administration" and "mock treated" by vehicle-treated.
Author Response
Reviewer 3: This interesting manuscript by Gaete and coworkers reviews the potential pros and cons of HIF-prolyl hydroxylases (PHDs) inhibitors as anti-tumor agents. Since PHDs are negative regulators of hypoxia-induced transcription factors, HIF-1a and HIF-2a, the review revolves around the effect of hypoxia on different activities associated with oncogenesis, such as angiogenesis, metabolism, inflammation, resistance to treatment and cancer stem cells. Afterwards, the influence of hypoxia on pheochromocytomas and paragangliomas is reviewed. Finally, an ambiguous section on the role of the PHD-HIF axis in tumor development is included as a preamble on a brief review of the four pharmacological PHD inhibitors tested in clinical trials in humans (although its effects on oncogenesis were obtained in preclinical models). Overall the review is well written and fairly comprehensive. However, perhaps the most important aspect of the potential role of PHDs is that they can have both anti-tumor and pro-tumor effects. This concept is only clearly evident in the section on the role of the PHD-HIF axis in tumor development. Perhaps the authors could consider moving that section to the beginning of the review.
We appreciate the stimulating words of the reviewer. We agree with the reviewer and moved this particular part to the beginning of the review. We completely underscore that this part puts the true role of PHDs in tumor development in the right context.
Major points.
- Many preclinical data suggest that the anti-tumor effect associated with PHD inhibition is a consequence of the "normalization" of the tumor vasculature, which promotes a better function of the blood vessels supplying O2 to the tumor and the reduction of hypoxia. The paradox of why the inhibition of PHDs, enzymes activated by the increase in pO2, improves vascular function is not sufficiently explained in the review. It is simply mentioned (page 3 lines 93-94) the study by Mazzone et al, in which the normalization of the vasculature of tumors implanted in hemizogoto mice for PHD-2 is observed, but this review does not go into the molecular mechanisms of this normalization. In fact, both the study by Mazzone et al, and others published more recently (see PMID: 29422508) indicate that the mechanism behind this normalization is the differential activity of PHDs on HIF-1a and HIF-2a hydroxylation. The low levels of PHD2, in the case of the hemyzogous mouse, or the partial inhibition of PHDs by nitric oxide (PMID: 29422508 and PMID: 17060326) allows the degradation of HIF-1a but the accumulation of HIF-2a, which seems to be a critical transcription factor in the remodeling of blood and lymphatic vessels. Furthermore, the stabilization of HIF-2a has also been associated with greater infiltration of cytotoxic lymphocytes and longer disease-free time in patients with colorectal cancer (PMID: 32591431). These molecular aspects must be discussed in depth so that the reader understands why PHD blockade may exert anti-tumor effects in vivo in some cases.
We thank the reviewer for this comment and giving us the opportunity to re-formulate this part of the review. We have used his/her suggestions and integrated them in our chapter on tumor micro-environment.
- Page 4, lines 149-151. From my point of view this sentence should be rewritten, since it seems to indicate that PHD2-KO macrophages do not express any polarization marker.
We reformulated this sentence.
- However, in the study by Guentsch et al. what is observed is that there are no differences in the expression of M1 and M2 markers between macrophages WT or PHD2-KO, both under resting conditions (bone marrow isolated) and after stimulation with polarizing agents. It is also important here that the authors discuss in some detail some surprising data obtained on PHD2-KO macrophages. Specifically, the fact that these PHD2-KO macrophages have affected the phagocytosis despite being highly glycolytic, a metabolic process associated with enhanced phagocytosis and polarization towards M1 macrophages (PMID: 31333642). Perhaps this indicates that it is not so important the metabolic reprogramming itself, but rather having an active mitochondrial program that provides enough energy for the phagocytic process.
We completely agree with the reviewer that the importance of metabolic reprogramming in these macrophages is not necessarily central to its phagocytotic activity. We have integrated your suggestion in the text
Minor points:
- Throughout the manuscript. I would appreciate the addition of a hyphen in HIF-1alpha and HIF-2alpha, for consistency with all the literature on this topic.
We have changed this throughout the manuscript
- 3, line 119. It seems that a reference is missing at the end of that sentence.
We corrected this.
- 4, line 131. I think the appropriate expression is Regulatory T-cells
We corrected this.
- 8, lines 344-345. The phrase should be rewritten by substituting "oral application" by "oral administration" and "mock treated" by vehicle-treated.
We corrected this.